# Production and Characterization of Peptide Antibodies to the C-Terminal of Frameshifted Calreticulin Associated with Myeloproliferative Diseases

**DOI:** 10.3390/ijms23126803

**Published:** 2022-06-18

**Authors:** Farah Perveen Mughal, Ann Christina Bergmann, Ha Uyen Buu Huynh, Sarah Hyllekvist Jørgensen, Inaam Mansha, Meliha Kesmez, Patrick Mark Schürch, Alexandre Pierre André Theocharides, Paul Robert Hansen, Tina Friis, Morten Orebo Holmström, Evaldas Ciplys, Rimantas Slibinskas, Peter Højrup, Gunnar Houen, Nicole Hartwig Trier

**Affiliations:** 1Department of Drug Design and Pharmacology, University of Copenhagen, Universitetsparken 2, 2100 Copenhagen Ø, Denmark; fp_mughal@yahoo.dk (F.P.M.); prh@sund.ku.dk (P.R.H.); 2Department of Autoimmunology and Biomarkers, Statens Serum Institut, Artillerivej 5, 2300 Copenhagen S, Denmark; ann_bergmann@hotmail.com (A.C.B.); uyenha_huynh@hotmail.com (H.U.B.H.); inaam.ihm@gmail.com (I.M.); tfs@ssi.dk (T.F.); 3Department of Biochemistry and Molecular Biology, University of Southern Denmark, Campusvej 55, 5230 Odense M, Denmark; sjzq@novonordisk.com (S.H.J.); melihakesmez@hotmail.com (M.K.); php@bmb.sdu.dk (P.H.); 4Global Research Technologies, Novo Nordisk A/S, Novo Nordisk Park, 2760 Måløv, Denmark; 5Department of Medical Oncology and Hematology, University Hospital Zurich, Rämistrasse 100, 8091 Zurich, Switzerland; pm.schuerch@gmail.com (P.M.S.); alexandre.theocharides@usz.ch (A.P.A.T.); 6National Center for Cancer Immune Therapy (CCIT-DK), Department of Oncology, Copenhagen University Hospital, Herlev Ringvej 75, 2730 Herlev, Denmark; morten.orebo.holmstroem@regionh.dk; 7Institute of Biotechnology, Life Sciences Center, Vilnius University, Saulėtekio al. 7, LT-10257 Vilnius, Lithuania; evaldas.ciplys@bti.vu.lt (E.C.); rimantas.slibinskas@bti.vu.lt (R.S.); 8Department of Neurology, Rigshospitalet Glostrup, Valdemar Hansens vej 13, 2600 Glostrup, Denmark

**Keywords:** calreticulin, epitope mapping, myeloproliferative neoplasms, peptide antibodies, frameshift mutations

## Abstract

Myeloproliferative Neoplasms (MPNs) constitute a group of rare blood cancers that are characterized by mutations in bone marrow stem cells leading to the overproduction of erythrocytes, leukocytes, and thrombocytes. Mutations in *calreticulin* (*CRT*) genes may initiate MPNs, causing a novel variable polybasic stretch terminating in a common C-terminal sequence in the frameshifted CRT (CRTfs) proteins. Peptide antibodies to the mutated C-terminal are important reagents for research in the molecular mechanisms of MPNs and for the development of new diagnostic assays and therapies. In this study, eight peptide antibodies targeting the C-terminal of CRTfs were produced and characterised by modified enzyme-linked immunosorbent assays using resin-bound peptides. The antibodies reacted to two epitopes: CREACLQGWTE for SSI-HYB 385-01, 385-02, 385-03, 385-04, 385-07, 385-08, and 385-09 and CLQGWT for SSI-HYB 385-06. For the majority of antibodies, the residues Cys^1^, Trp^9^, and Glu^11^ were essential for reactivity. SSI-HYB 385-06, with the highest affinity, recognised recombinant CRTfs produced in yeast and the MARIMO cell line expressing CRTfs when examined in Western immunoblotting. Moreover, SSI-HYB 385-06 occasionally reacted to CRTfs from MPN patients when analysed by flow cytometry. The characterized antibodies may be used to understand the role of CRTfs in the pathogenesis of MPNs and to design and develop new diagnostic assays and therapeutic targets.

## 1. Introduction

Myeloproliferative neoplasms (MPNs) are a group of haematological malignancies characterised by excessive proliferation of hemopoietic lineage cells, e.g., megakaryocytes, neutrophilic granulocytes, and red blood cells [1,2,3,4]. Depending on the clinical phenotype and paraclinical parameters, these diseases are classified as polycythaemia vera (PV), essential thrombocythemia (ET), and myelofibrosis (MF) [1,2,3,4]. MPNs are initiated when genetic changes in a hematopoietic stem cell (HSC) lead to clonal expansion, resulting in malignant clones [5]. The MPN-related mutations may occur at different stages during the differentiation of HSCs to blood cells, resulting in the overproduction of a single or multiple blood lineages [6,7].

MPNs are typically associated with mutations in the Janus kinase (JAK), myeloproliferative leukaemia virus oncogene (MPL), and calreticulin (CRT). Most common are JAK mutations, followed by MPL and CRT mutations [3,4,8,9].

The MPN-associated CRT mutations, mainly insertions and deletions (indels), were originally identified by exome sequencing [10,11] and are found in exon 9 of the *CRT* gene. These indels result in frameshifts, which yield very similar alterations in the C-terminus of the CRT protein, seen as a variable stretch of basic amino acids and ending in a common C-terminal sequence without the endoplasmic reticulum (ER) retention signal KDEL found in the wild-type (wt) CRT protein [1,2,3,4,8,9,10,11,12] (Figure 1). To date, more than 50 different types of *CRT* gene mutations have been reported. The mutations are found in approximately 20–25% of ET and 25–30% of MF and are rarely associated with PV [11,12].

The current therapies for MPNs only prevent disease progression and prolong life. Hence, the available therapies for PV and ET, including low-dose aspirin, phlebotomy, and hydroxyurea, are focused on symptom management, notably thrombosis and haemorrhage [14]. Despite our knowledge of the genetic basis of these diseases, MPN therapies have not yet exploited the knowledge of mutations in the treatment of MPNs, which could selectively target the malignant clones [14]. JAK inhibitors and disease modifiers currently being tested in clinical trials have shown potential to reduce disease-related symptoms and may even improve survival in MPN subgroups [15,16], although none of them are capable of eradicating the malignant clone. In general, the treatment of MPNs using disease modifiers is further complicated, as the specific treatment depends on the individual mutation [16]. Diagnosis of MPNs is aided by the detection of the various associated mutations by polymerase chain reaction, sequencing, or other DNA-based methods [17,18,19,20,21,22,23,24,25,26,27,28,29,30,31], but the detection of disease-associated mutations may be facilitated or aided by mutation-specific immunohistochemistry (IHC) [31,32,33], methods which supplement each other. Moreover, antibodies such as these are invaluable reagents for studying the properties of CRT in relation to MPN. A monoclonal antibody (mAb) against frameshifted CRT (CRTfs) and its reaction with mutation-harbouring cells have been described [34]. This antibody was produced by immunisation with a mutation-specific 22-amino acid residue C-terminal peptide of CRTfs and screened by IHC assays of HEK293 cells transfected with a plasmid coding for the same peptide fused to the C-terminus of enhanced green-fluorescent protein. Although commercially available, this antibody is limited to CRTfs detection in paraffin-embedded fixed cell tissue; thus, more antibodies for the detection of CRTfs and as potential therapeutics are needed.

Due to the importance of CRTfs mutations in MPNs, we have produced several mAbs to the C-terminus of CRTfs and characterized their epitope specificity and properties in relation to an enzyme-linked immunosorbent assay (ELISA). Moreover, a selected antibody was tested for reactivity to the MARIMO cell line expressing CRTfs L367 in Western immunoblotting and flow cytometry (FC) as well.

## 2. Results

### 2.1. Testing of Mouse Immune Responses to Peptide Vaccines

Initially, mice were immunised with three peptides from the C-terminal end of CRTfs, whereafter samples from the fourth bleed were tested for reactivity to biotinylated peptides in ELISA to determine whether immunisations resulted in the production of peptide-specific CRTfs antibodies (Figure 2).

No notable antibody reactivity to all peptides tested was detected in bleeds from mice immunised with peptide 1, reflecting a poor immune response to the peptide (Figure 2a). Bleeds from mice immunised with peptide 2 and 3 only reacted with peptides 1 and 2 (Figure 2b,c). Interestingly, none of the samples from mice immunised with peptides 2 and 3 reacted to peptide 3, which may be ascribed to sterical hindrance from biotin or the peptide being too short to fold into a recognizable conformation during testing.

Based on these findings, mice with the highest reactivities were selected for fusion, and antibody clones were selected for their reactivity to peptide 2. 

### 2.2. Reactivity of Cell-Culture Supernatants to Frameshifted Calreticulin

In total, eight clones were generated (SSI-HYB 385-01, 385-02, 385-03, 385-04, 385-06, 385-07, 385-08, 385-09) (Table 1), which were tested for reactivity to biotinylated peptide 2 (CREACLQGWTE) in a dilution ELISA (Figure 3).

The majority of the cell-culture supernatants contained similar antibody levels in the starting dilution (1:10), except for SSI-HYB 385-09, where the antibody levels were elevated compared to the remaining cultures. Titres yielding half-maximal reactivity in ELISA for SSI-HYB 385-08 were 1:80; 1:160 for SSI-HYB 385-07; 1:320 for SSI-HYB 385-03, SSI-HYB 385-04, and SSI-HYB 385-09; 1:640 for SSI-HYB 385-01 and SSI-HYB 385-02; and finally, 1:2560 for SSI-HYB 385-06, indicating that the affinity of SSI-HYB 385-06 was notably higher compared to the remaining HYBs.

Next, the reactivity of the generated antibodies to CRTfs protein was analysed to determine whether the HYBs recognized the CRTfs K385 mutant. As presented in Figure 4, mainly SSI-HYB 385-06 reacted with the frameshifted protein, and the reactivity was found to be specific, as no reactivity was observed to the CRTwt. Moreover, the SSI-HYBs 385-07 and 385-08 showed weak reactivity to CRTfs K385. 

### 2.3. Identification of the N- and C-Terminal Amino Acids Essential for Antibody Reactivity

To identify the terminal amino acids essential for antibody binding, the reactivity of the SSI-HYBs to systematically N- and C-terminally truncated resin-bound peptides was tested by modified ELISA (Figure 5). Peptide 2 (CREACLQGWTEA) was used as template for the generation of N- and C-terminally truncated peptides.

The majority of the HYBs only reacted with the CREACLQGWTEA peptide, whereas a few clones reacted to the N-terminally truncated peptide CLQGWTEA (SSI-HYB 385-03, SSI-HYB 385-06, SSI-HYB 385-09) (Figure 5a). Except for SSI-HYB 385-06, none of the HYBs reacted with the C-terminally truncated peptides (Figure 5b). SSI-HYB 385-06 was the only clone that reacted to the N-terminally truncated peptides CLQGWTEA, ACLQGWTEA, ACLQGWTEA, EACLQGWTEA, and REACLQGWTEA. Moreover, SSI-HYB 385-06 reacted to the C-terminally truncated peptide CREACLQWT.

Collectively, these findings indicate that the complete peptide CREACLQGWTEA is essential for antibody binding for the majority of the clones tested, either through direct interactions or by contributing to a correct conformation for antibody reactivity, whereas the epitope of SSI-HYB 385-06 was located within the CLQGWTEA peptide, where the N-terminal Cys and C-terminal Thr residues identified the terminal borders essential for antibody reactivity. These findings indicate that the amino acid sequence CLQGWT may constitute the minimum functional epitope of SSI-HYB 385-06, although verification is necessary.

### 2.4. Importance of Cysteine Residues for Antibody Binding

Preliminary findings indicated that the N-terminal Cys^1^ residue was important for antibody binding. To confirm this result and to examine the role of the two Cys residues in the peptide, substituted peptides [C→A]^1^, [C→A]^5^, and [C→A]^1,5^ were tested for antibody reactivity by modified ELISA.

All SSI-HYBs bound to the [C→A]^5^ substituted peptide, whereas SSI-HYB 385-06 was the only antibody to react with the [C→A]^1^ and [C→A]^1,5^ substituted peptides, confirming that the N-terminal Cys^1^ was essential for antibody binding for the remaining SSI-HYBs (Figure 6).

Based on the current findings, Cys^5^ was substituted with Ala in future experiments to avoid possible disulphide bond formation, which may interfere with antibody reactivity. 

### 2.5. Identification of Essential Amino Acids for Antibody Binding by Screening of Ala- and Functionality-Substituted Peptides

To identify amino acids essential for binding, antibody reactivity to Ala-substituted peptides and functionality-substituted peptides was analysed by modified ELISA. The peptide CREAALQGWTEA was used as the template for the generation of Ala-substituted peptides.

As presented in Table 2, antibody reactivity was notably reduced when Leu^6^ and Trp^9^ were substituted with Ala, indicating that these positions are essential for reactivity. None of the HYBs tolerated [W→A]^9^ substitution, whereas only two HYBs (SSI-HYB 385-01, 385-06) tolerated [L→A]^6^ substitution. However, all HYBs tolerated [L→I]^6^ substitution, and all HYBs except for SSI-HYB 385-02 and SSI-HYB 385-04 tolerated [W→F]^9^ substitution (Table 3), indicating that Leu^6^ and Trp^9^ mainly contribute with side-chain functionality rather than side-chain specificity for the majority of the HYBs. Moreover, all HYBs tolerated [R→A]^2^, [Q→A]^7^, [A→G]^4^, and [A→G]^12^ substitutions, indicating that these residues do not contribute directly to the antibody–antigen interface. Except for SSI-HYB 385-06, all SSI-HYBs were intolerant to [C→M]^1^ substitution, confirming that the free -SH group at the N-terminal end of the peptide is crucial for antibody binding, as shown earlier.

SSI-HYB 385-01 was intolerant to [W→A]^9^, [T→A]^10^, [E→A]^11^, and [E→D]^11^ substitutions, whereas [W→F]^9^ and [T→S]^10^ substitutions were tolerated, indicating that an aromatic side chain in position 9, a hydroxyl side chain group in position 10, and the specific side chain of Glu^11^ were crucial for antibody binding.

SSI-HYB 385-02 was very sensitive to Ala substitutions, as five substitutions ([E→A]^3^, [L→A]^6^, [G→A]^8^, [W→A]^9^, [T→A]^10^) notably reduced antibody reactivity. However, SSI-HYB 385-02 was tolerant to the majority of functionality substitutions, except for [W→F]^9^, indicating that a carboxylic acid in position 3, a hydrophobic amino acid in position 6, a small polar amino acid in position 8, Trp^9^, and finally, a hydroxyl group in position 10 were essential for SSI-HYB 385-02 reactivity.

Similar to SSI-HYB 385-02, SSI-HYB 385-03 and SSI-HYB 385-04 were intolerant to [L→A]^6^, [G→A]^8^, [W→A]^9^, and [T→A]^10^ substitutions, although SSI-HYB 385-03 reacted to all of the functionality-substituted peptides, whereas SSI-HYB 385-04 was intolerant to the [W→F]^9^ substitution. Collectively, these findings indicated that SSI-HYB 385-03 primarily depended on the functional side-chain reactivity in the critical positions rather than a specific amino acid side chain, whereas HYB 385-04 depended on the specific side chain of Trp^9^ in combination with side-chain functionality in the remaining positions. Of note, SSI-HYB 385-02, SSI-HYB 385-03, and SSI-HYB 385-04 were intolerant to the [G→A]^8^ substitution; however, no further substitutions were introduced in this position, as Gly contains a hydrogen atom in the side chain. A substitution of Gly with Ala, which contains a –CH_3_ group in the side chain, is the closest possible functionality substitution.

SSI-HYB 385-06 reacted to all of the Ala-substituted peptides, except [W→A]^11^. However, as the HYB was tolerant to all functionality substitutions introduced, the results indicated that SSI-HYB 385-06 primarily depended on an aromatic side chain in position 9 in combination with the peptide backbone for antibody reactivity.

In general, the SSI-HYBs were tolerant to the introduced functionality substitutions with a few exceptions. SSI-HYBs 385-07, 385-08, and 385-09 did not react to [L→A]^6^, [W→A]^9^, or [E→A]^11^ substituted peptides, nor to the [E→D]^11^ substituted peptide, but they were tolerant to the remaining functionality substitutions. Collectively, these findings indicated that the epitope of the three HYBs was identical and that amino acids with similar functionalities were tolerated in positions 6 and 9, whereas the negatively charged Glu^11^ was crucial for antibody binding.

As mentioned earlier, the SSI-HYB reactivities were not influenced by the [A→G]^12^ substitution, indicating that Ala^12^ was not crucial for antibody binding. The same applies to the [E→A]^11^ substitution for SSI-HYBs 385-02, 385-03, 385-04, and 385-06 and to the [T→A]^10^ substitution for SSI-HYB 386-05, although screening of C-terminally truncated peptides revealed that these amino acids are essential for antibody binding. Although not contributing directly to the antibody–antigen interface, these amino acids appeared to be essential for the stabilization of the peptide structure and, thereby, for locking the critical residues into a correct orientation for a stable antibody–antigen interaction.

Based on these findings, the following minimum functional epitopes were identified for the SSI-HYBs: CREACLQWTE for SSI-HYB 385-01, SSI-HYB 385-02, SSI-HYB 385-03, SSI-HYB 385-04, SSI-HYB 385-07, SSI-HYB 385-08, and SSI-HYB 385-09; and finally, CLQGWT for SSI-HYB 385-06.

### 2.6. Structural Epitope Analysis

Following epitope identification, the secondary structure of the proposed epitope CREACLQGWTE was predicted using the PEP FOLD server.

As depicted in Figure 7a,b, the peptide folded into an α-helical structure. The α-helix structure results from hydrogen bonding between the -NH and -C=O groups in the peptide backbone without involving the amino acid side chains; hence, the amino acid side chains protrude from the α-helix, making them accessible for binding to the antibody, as illustrated. As seen, the two -SH groups (yellow) are in proximity to each other, making an intramolecular disulphide bond formation possible.

The general characteristics and possible side-chain interactions of helices are easier visualized if the sequence is plotted in a two-dimensional helical wheel, where the amino acid side chains are projected onto a plane perpendicular to the axis of the helix, as presented in Figure 7c [35].

As seen, the amino acid side chains, which are crucial for antibody binding, are located on the same side of the helix wheel: Cys^1^ and Glu^11^ for SSI-HYB 385-01, SSI-HYB 385-07, SSI-HYB 385-08, and SSI-HYB 385-09 and Cys^1^ and Trp^9^ for SSI-HYBs 385-02 and SSI-HYB 385-04. 

The presence of hydrophobic residues is thought to stabilize the structure by intrahelical interactions, which constitute the internal “core” of the helix. The hydrophobic residues tend to be directed inside the helix, while the polar residues are located on the surface [35]. As seen, two pairs of more or less hydrophobic amino acids were found on the same side of the wheel: (1) Ala^4^ and Gly^8^ and (2) Leu^6^ and Trp^9^. Consistently, Ala^4^ and Gly^8^ were hidden inside the helix in the predicted three-dimensional structure. Furthermore, the amino acid contacts found in the three-dimensional structure of the epitopes (Leu^6^, Thr^10^ and Gln^7^, Thr^10^) were located very close in the helical wheel, indicating that these residues stabilize the helical structure.

### 2.7. Reactivity of SSI-HYB 385-06 in Western Blotting

Initial screenings indicated that SSI-HYB 385-06 exhibited a high affinity and reactivity to full-length CRTfs K385 and peptide 2, and based on this, SSI-HYB 385-06 was purified and characterized further in Western immunoblotting.

As presented in Figure 8, SDS-PAGE analysis revealed two bands at 43 kDa and 95 kDa for CRTfs K385 under non-reducing conditions (monomer, dimer) and a single band at 40 kDa under reducing conditions. For CRTfs L367, single bands at 35 kDa and 38 kDa were seen under non-reducing and reducing conditions, respectively, and for CRT, major bands at 46 kDa and 49 kDa and minor bands at 40 kDa and 43 kDa were seen under non-reducing and reducing conditions, respectively.

Coomassie Brilliant Blue staining (Figure 8a) depicted bands corresponding to monomer and dimer for CRTfs K385, representing oligomeric forms under non-reducing conditions, whereas under reducing conditions, a single band at 60 kDa was seen. Similar bands for CRT L367 and Rec CRT were identified.

Immunoblotting with SSI-HYB 385-06 (Figure 8b) showed bands corresponding to monomer and dimer for CRTfs K385, as well as a smear at approximately 240–350 kDa, representing oligomeric forms under non-reducing conditions, whereas under reducing conditions, a single band at 60 kDa was seen. Very weak bands were seen for CRTfs L367, whereas no bands were visualised for Recombinant CRT. When the same blot was further developed with a mAb recognizing the N-terminal of CRT (CRT FMC 75 mAb) (Figure 8c), additional bands at 46 kDa were seen for CRTfs K385 under both non-reducing and reducing conditions. For CRTfs L367, an additional band at 38 kDa was seen under non-reducing conditions, and a band at 43 kDa appeared under reducing conditions. For CRT, bands corresponding to the major and minor bands seen by SDS-PAGE appeared. Based on the current findings, these results indicated that a major part of CRTfs K385 proteins were cleaved in the C-terminus. However, due to disulphide bond formation between the protein and the cleaved C-terminal peptide, the cleaved peptide remained attached to the protein, whereas the cleaved C-terminal peptide was absent from the CRTfs L367 protein, as presented with a lower molecular weight. These results were confirmed by mass spectrometry, which showed cleavage of the CRTfs proteins at positions 281 and 370/371, respectively, for a major part of CRTfs K385 and CRTfs L367 (Appendix A).

Finally, we tested the reactivity of SSI-HYB 385-06 to MARIMO cells by Western blotting (Figure 9). This cell line is negative for JAK and MPL mutations but carries a heterozygous 61-basepair deletion (c.1099_1159del; L367fs *43), which, like all other reported CRT mutations, results in a + 1-basepair shift in the reading frame, leading to a novel C-terminus [36].

Figure 9 illustrates the reactivity of SSI-HYB 385-06 to MARIMO cells +/− brefeldin A (BFA) treatment analysed by Western blotting. BFA indirectly inhibits protein transport from the ER to the Golgi complex. As presented, no CRTfs was detected intracellularly in the absence of BFA (−BFA) as compared to total CRT (Figure 9a). In contrast, high levels of endogenous CRTfs were detected in BFA-treated MARIMO cells, indicating that synthesized CRTfs easily leaves the ER and only accumulates intracellularly upon BFA treatment, in contrast to CRTwt, which primarily remains in the ER. The current findings confirm that SSI-HYB 385-06 recognizes human CRTfs.

### 2.8. Reactivity of SSI-HYB 385-06 to Frameshifted Calreticulin in Flow Cytometry

Besides being tested in ELISA and Western blotting, it was determined whether the SSI-HYB 385-06 could be used for the detection of CRTfs using FC. Approximately 50% of the MPN patients were positive for mutations in CRT. Accordingly, patients positive for CRT mutations were analysed for reactivity with SSI-HYB 385-06. 

Figure 10 illustrates the reactivity of SSI-HYB 385-06 and an isotype control to white blood cells from a CRTfs-positive MPN patient. 

Using the SSI-HYB 385-06 clone, intracellular staining of white blood cells was observed, confirming that SSI-HYB 385-06 functions in FC as well. Moreover, these findings are in accordance with the BFA treatment of MARIMO cells, illustrating that CRTfs is present endogenously. A similar extracellular staining pattern was observed for some MPN patients, although the reactivity was low (results not shown). This is in agreement with previous results, where patients with MPN have presented with elevated CRTfs levels on the surface of granulocytic cells [32,33,34].

## 3. Discussion

MPNs are rare hemopoietic malignancies that have been shown to have increasing incidence rates [1,2,3,4]. In this study, eight antibodies to CRTfs related to MPNs were characterized. The minimum functional epitopes were identified as CREACLQGWTE for SSI-HYBs 385-01, SSI-HYBs 385-02, 385-03, 385-04, 385-07, 385-08, and 385-09 and CLQGWT for SSI-HYB 385-06. The results obtained may be used to detect the presence of CRT mutants in MPNs and may contribute to the development of new therapeutics to selectively target the malignant clones.

All antibodies recognised the CREACLQGWTEA peptide from the C-terminal of CRTfs and were characterized using synthetic resin-bound peptides, which provide an efficient and rapid screening tool for antibody characterization [38,39]. Resin-bound peptides are advantageous, as they are linked to a solid support by only one amino acid, allowing the remaining amino acids to achieve a conformational structure. In contrast, when free peptides are coated directly to the solid surface of microtiter plates, their conformation may change, and specific contact amino acids may bind non-specifically to the wells, masking critical contact residues for antibody binding [40,41].

Initially, N- and C-terminally truncated peptides were used to identify the terminal borders of the binding sites. However, only limited reactivity was observed to the truncated peptides, indicating that the majority of the SSI-HYBs depended on the complete peptide for antibody binding, whereas the sequence CLQGWT presumably constituted the minimum functional epitope of SSI-HYB 385-06, although this remains to be verified.

Using substituted peptides, the role of amino acid side chains for antibody binding within epitopes can be determined [42,43], hence, Ala-substituted peptides were screened for reactivity, where each amino acid was replaced one-by-one with the non-bulky and chemically inert Ala residue in the CREAALQGWTEA peptide. Cys^5^ was substituted by Ala to avoid unwanted disulphide bond formation, as preliminary findings indicated that none of the HYBs depended on Cys^5^ for reactivity. In contrast, all SSI-HYBs depended on Cys^1^ for reactivity, except for SSI-HYB 385-06. These findings were confirmed when screening antibody reactivity to the functionality-substituted peptide [C→M]^1^, as the substitution notably reduced antibody binding for the C^1-^dependent SSI-HYBs. Moreover, these results indicate that Cys^5^ was favoured during coupling, which also explains why a free -SH group rather than a disulphide bond is favoured for reactivity.

Four sets of important contact residues were identified: Cys^1^, Leu^6^, Gly^8^, Trp^9^, and Thr^10^ for SSI-HYBs 385-02, 385-03, and 385-04 (and Glu^3^ for SSI-HYB 385-02); Cys^1^, Leu^6^, Trp^9^, and Glu^11^ for SSI-HYBs 385-07, 387-08, and 385-09; Cys^1^, Trp^9^, Thr^10^, and Glu^11^ for SSI-HYB 385-01; and Trp^9^ for SSI-HYB 385-06. In general, SSI-HYB 385-06 differed compared to the remaining HYBs, since its binding was exclusively affected by the [W→A]^9^ substitution.

Ala scans were supplemented by functionality scans, which were applied to determine the specific contributions of each amino acid. Based on this, Cys^1^, Leu^6^, Gly^8^, Trp^9^, Thr^10^, and Glu^11^ were substituted with amino acids of similar functionality (Met, Ile, Ala, Phe, Ser, and Asp, respectively). Collectively, the side chains of amino acids Cys^1^, Trp^9^, and Glu^11^ contributed directly to the antibody–antigen interface, whereas the remaining substituted amino acids contribute to the interface through the amino acid side-chain functionality.

The contribution of the essential amino acids differed. Thus, Trp was involved in binding through its aromatic structure. In contrast, Cys contains a -SH group in its side chain, which forms a covalent disulphide (S-S) bond with adjacent (primary sequence or folded structure) Cys residues [44]. Leu is an aliphatic non-polar amino acid, which participates in Van der Waals interactions, whereas Gly only contains a hydrogen atom in the side chain and mainly only contributes to backbone interactions and introduces conformational flexibility [42]. Glu contains a carboxylic acid in its side chain, which can form ionic bonds and functions as a hydrogen bond acceptor. Thus, the critical residues interact with the SSI-HYBs through ionic bonds, hydrogen bonds, and Van der Waals interactions. 

The importance of hydrophilic amino acids and charged amino acids, but also hydrophobic amino acids, for antibody reactivity is in accordance with the literature, describing that epitope structures often are rich in hydrophilic and charged amino acids and occassionally hydrophobic amino acids as well [43,45,46]. In a recent study, the epitopes within glutamic acid decarboxylase 65 were identified, which revealed that the reactivity of these antibodies depended on the amino acids Phe, Trp, Thr, and Leu [43], similar to the SSI-HYBs, indicating that these residues are common critical residues. Moreover, another recent study described the importance of charged residues as crucial for antibody reactivity [39,45,47], which is also in accordance with findings within this study. In the study by Bergmann and colleagues, two mAbs raised to the N-terminal (amino acids 34–41) and the C-terminal (amino acids 362–373) of CRTwt were characterized. In these epitopes, charged amino acids in combination with the peptide backbone were found to be important for reactivity [37]. This is in accordance with results obtained in the current study, where among others, a crucial Glu in combination with a peptide backbone was essential for reactivity.

Primarily, SSI-HYB 385-06 appeared to depend on backbone reactivity, although the [W→A]^9^ substitution reduced antibody reactivity. However, as the [W→F]^9^ substitution did not reduce the reactivity of SSI-HYB 385-06 to the substituted peptide, the current findings indicate that the SSI-HYB mainly depended on the peptide backbone for reactivity. The components of the side chain define the physicochemical character of an amino acid. If certain amino acids only interact with an antibody through their backbones, there may be no real physicochemical preference for them over other amino acids [45]. This may indicate that the side chains of Trp^9^ and Phe did not contribute to a direct interaction with SSI-HYB 385-06 but interacted with the peptide backbone that caused a favourable conformation recognized by the antibody. The backbone of a polypeptide chain is rich in hydrogen bonding potential, as each amino group in the polypeptide chain contributes with one carbonyl group (hydrogen bond acceptor) except for proline and one NH group (hydrogen bond donor). These groups interact with each other and with side-chain functional groups to stabilize the secondary and tertiary structures. This remains to be verified.

Several backbone-dependent antibodies recognizing relative short epitopes similar to HYB 385-06 have been described in the literature [39,48,49]. Recently, a mAb to the N-methyl D-Aspartate receptor NMDAR1 was characterized, which was found to depend on a positively charged amino acid combined with noncovalent backbone interactions. Here, the amino acids in a minimum functional epitope could be replaced by amino acids of similar functionality or different side-chain functionality without compromising the antibody reactivity, indicating that backbone conformation and side-chain functionality rather than specific side-chain reactivity were necessary for antibody reactivity [39], similar to SSI-HYB 385-06.

Based on truncation and substitution experiments, the epitopes identified were 10–11 amino acids long, except for the SSI-HYB 385-06 epitope, which was identified as the CLQGWT epitope. However, as seen, SSI-HYB 385-06 was tolerant to the [C→A]^5^ substitution, indicating that the contribution of Cys^5^ is related to the structure rather than the actual reactivity to the peptide. The same applies to the C-terminal reactivity of the remaining SSI-HYBs. All HYBs tolerated the [A→G]^12^ and some the [E→A]^11^ substitution, indicating that the amino acids do not contribute to the antibody–antigen interface; rather, the amino acids contribute to stabilizing the α-helix structure. As presented, the peptides may fold into an α-helical structure. The α-helix structure results from hydrogen bonding between the -NH and -C=O groups in the backbone of the peptides without involving the amino acid side chains. Although Cys^1^ and Cys^5^ are in proximity to each other, the residues are believed to make contact to other Cys residues in the native CRT protein rather than to each other based on the orientation of the Cys residues relative to each other. This remains to be confirmed. As nicely depicted in the three-dimensional structures, the critical amino acid side chains are accessible for antibody binding.

SSI-HYB 385-06 obtained the highest affinity and was the only antibody that showed high reactivity to CRTfs K385 in ELISA, although SSI-HYB 385-07 and 385-08 showed weak reactivity as well. Antibody reactivity to the denatured form of CRTfs K385 was only analysed for SSI-HYB 385-06; however, it would be interesting to see whether the remaining SSI-HYBs bind to the denatured form, as no reactivity was found to the native form. Moreover, SSI-HYB 385-06 was tested in Western blotting, where the antibody bound to the protein in the blot and discriminated between CRT forms with and without the C-terminal peptide attached. In addition, SSI-HYB 385-06 reacted with endogenous CRFfs in MARIMO cells and was capable of detecting intracellular and/or extracellular CRTfs in the white blood cells of some MPN patients, although the reactivity was generally low. This is in agreement with previous results, where patients with MPN have presented with elevated CRTfs levels on the surface of granulocytic cells [32,33,34] and illustrates the wide application of SSI-HYB 385-06 in several assays.

Peptide antibodies are applied in a variety of immunoassays, including immunoprecipitation, immunoblotting, and IHC [50,51], where mutation-specific peptide antibodies among others can be used in the diagnosis of different cancers [50,51]. In addition, they are used for microscopic examination of cancerous tissues, where the binding of an antibody to the final target is detected by fluorescence [50,51]. There is a notable interest in developing novel and improved antibody treatment strategies. However, challenges such as stability, bioavailability, and immunological engagement are faced in the design, manufacture, and formulation of therapeutic antibodies. Here, we characterized eight antibodies, which, in theory, may function as therapeutics in CRTfs-specific MPNs and may be used for diagnostic purposes.

Recent advances in the diagnostics of CRTfs-specific MPNs relate to primary myelofibrosis (PMF) and ET, which lack mutations in JAK2 and MPL. CRTfs mutations are momentarily only detectable in molecular assays. However, due to a significant CRT mutation heterogeneity, a simple assay is needed, which preferably detects the very C-terminal, where no substitutions currently have been detected. A single mAb (CAL2) directed to the C-neoterminus has been generated. Although the antibody detected CRTfs in IHC assays, the application of the antibody may be limited to detection in formalin-fixed and paraffin-embedded tissues [32]. Hence, new and well-characterized CRTfs antibodies are needed to optimize the detection of CRTfs in various assay formats. In addition, it remains to be determined whether the CAL2 antibody is able to access CRFfs when CRT is in a complex with MPL. Studies of truncated mutants have revealed that residues 376–383 from the C-terminus of CRT_Del52_ are required to activate MPL-mediated signalling [52]. Moreover, recent studies indicate that the C-terminal Cys residues are essential for reactivity as well, as mutations C400A/C404A in CRT_Del52_ reduced its ability to bind MPL [53]. Based on these findings, our current belief is that SSI-HYB 385-06, which does not have a dependency of any of the C-terminal Cys residues, would be able to interact with CRTfs in a complex with MPL. This was partly confirmed when analysing SSI-HYB 385-06 reactivity to MARIMO cells in Western blotting and to white blood cells by FC, as the antibody bound to CRFfs, which very well may be in a complex with MPL. This remains to be verified.

In these studies, SSI-HYB 385-06 directed to the C-terminus of CRTfs showed especially promising results. The proposed epitope of SSI-HYB 385-06 was identified in human NOTCH4 by pBLAST; however, since NOTCH4 has a molecular weight of approximately 200,000 Da and is a transmembrane protein with a tissue and cellular distribution, we do not believe that this will interfere with the use of this antibody, but it should be kept in mind when using this antibody. Moreover, the identified motif ^870^CLQGWT^870^ is proposed to be located in a flexible turn structure in NOTCH4 (identified using Alphafold), which is different from the proposed α-helix structure identified in CRTfs, supporting that SSI-HYB 385-06 may not interact with NOTCH4 due to structural differences. 

Collectively, the current findings may indicate that the SSI-HYB 385-06 clone has a wide application as a research tool, in diagnostics assays and as a potential therapeutic drug.

## 4. Materials and Methods

### 4.1. Reagents

Tris-Tween-NaCl (TTN) buffer (0.05 M Tris, 0.3 M NaCl, 1% Tween 20, pH 7.4), alkaline phosphatase (AP) substrate buffer (1 M diethanolamine, 0.5 mM MgCl_2_, pH 9.8), and carbonate buffer (0.05 M sodium carbonate, pH 9.6) were obtained from Statens Serum Institut (Copenhagen, Denmark). AP-conjugated goat anti-mouse IgG, AP substrate tablets *para*-nitrophenylphosphate (*p*NPP), streptavidin, trifluoroacetic acid (TFA), triisopropylsilane (TIS), Tentagel S RAM resin, dithioretiol (DTT), N,N-diisopropylethylamine (DIEA), thioanisole (THIO), 9-fluorenylmethoxycarbonyl (Fmoc)-L-amino acids, 5-bromo-4-chloro-3-indoyl phosphate (BCIP)/nitro-blue tetrazolium chloride (NTB) tablets, and iodoacetic acid-N-hydroxysuccininmidester (IAANHOSu) were obtained from Sigma Aldrich (St. Louis, MO, USA). 1-hydroxy-7-azabenzotriazole (HOAt) was obtained from ChemPep (Wellington, FL, USA). Sodium hydroxide, sodium azide, Tween 20, paraformaldehyde, and bovine serum albumin (BSA) were obtained from Merck Eurolab (Vaugereau, France). Piperidine (PIP), dimethylformamid (DMF), and diethylether were obtained from VWR Chemicals (Gliwice, Poland). Fmoc L-amino acids and HATU were obtained from Iris Biotech GMBH (Markredwitz, Germany). Fmoc-L-amino acids were obtained from Novabiochem (Darmstadt, Germany). CRT FMC 75 mAb (Cat.no. ADI-SPA-601) was purchased from Enzo Life Sciences (Farmingdale, NY, USA). Al(OH)_3_ (2%) was obtained from Brenntag Biosector (Frederikssund, Denmark). Precision Plus Protein Standard was purchased from BIO-RAD (Hercules, CA, USA). Coomassie brilliant blue, native PAGE running buffer (pH 7.5), 4–20% Tris-glycine gels, PVDF membranes, sample buffer (SB), Fluorescein-5-isothiocyanate (FITC)-labelled goat anti-mouse IgG and Roswell Park Memorial Institute (RPMI) medium were obtained from Thermo–Fisher (Waltman, MA, USA). MPN sera (n = 11) were obtained from the Department of the National Center for Cancer Immune Therapy, Department of Oncology, Copenhagen University Hospital. Initial peptides used for immunisation were purchased from Schäfer-N peptides (Lyngby, Denmark) (Table 4); the remaining resin-bound peptides used for characterisation were synthetized by traditional Fmoc-based solid-phase peptide synthesis (SPPS), as described in Section 4.3 [40]. Recombinant CRTwt and CRTfs L367 and K385 produced in yeast were obtained from Baltymas (Vilnius, Lithuania). The CRTfs proteins were produced and stability-tested, as previously described [54,55]. BFA was obtained from Invitrogen (Waltman, MA, USA). Peroxidase AffiniPure goat anti-rabbit IgG and goat anti-mouse IgG were obtained from Jackson ImmunoResearch (Cambridgeshire, United Kingdom). FACS Lysing solution was obtained from (BD Fastimmune, Franklin Lakes, NJ, USA).

### 4.2. Peptides

Synthetic peptides of varying lengths were used for generation and characterization of peptide antibodies to the C-terminal of CRTfs. Free peptides used for immunisation are listed in Table 4, whereas resin-bound peptides used for characterization are listed in Table 5.

### 4.3. Solid-Phase Peptide Synthesis of Resin-Bound Peptides

Resin-bound peptides were synthesized directly on resins by Fmoc-based SPPS using standard protocols [39,41]. Tentagel S RAM resin (50 mg, loading: 0.24 mmol/g) was used for synthesis. Fmoc-protected L-amino acids and coupling reagents (HOAt, HATU, and DIEA) were used in twofold excess. Coupling was performed for 2 h at room temperature (RT). After each coupling, resins were rinsed with DMF. Fmoc was removed using 20% *v*/*v* PIP in DMF for 20 min. Side-chain protecting groups were cleaved from the resin using a cocktail of 95% TFA, 2.5% THIO, and 2.5% TIS for 2 h at RT.

For generation of C-truncated resins, Ala-substituted peptides and functionality-substituted peptides were synthesized in individual syringes. For generation of N-truncated peptides, the first four amino acids were coupled, whereafter approximately 20 mg resin was removed from the syringe after coupling of each amino acid.

### 4.4. Amino Acid Analysis

Amino acid analysis was carried out essentially as described by Højrup [56]. Samples of 2–4 µg protein were dried in 0.5 mL polypropylene tubes, the lids were punctured, and the tubes were placed in 25 mL glass vials. Following this, 200 µL of 6N HCl, 0.1% phenol, and 0.1% 2-thioglycolic acid were added and the vial closed with a MinInert valve (VICI) after being flushed by argon and evacuated to <1 mBar. Following 20 h hydrolysis at 110 °C, samples were dried, re-dissolved, and analysed on a BioChrom 30+ amino acid analyser according to the manufacturer’s recommendations [57].

### 4.5. Mass Spectrometry

For the intact mass analysis of CRTfs K385 and CRTfs L367, 75 pmol protein was injected into the mass spectrometer. The analysis was performed on a nanoACQUITY ultra-performance liquid chromatography system online, coupled with a Waters Synapt G2 ESI quadrupole time-of-flight mass spectrometer.

The protein was desalted on a MassPREP Micro Desalting Column (1.0 mm × 50 mm) (Waters) with solvent A (0.23% FA) at a flow rate of 500 µL/min for 2 min. After desalting, the protein was eluted with the following 7 min gradient: 5–50% of solvent B (0.23% FA in acetonitrile) for 3 min, 50–90% solvent B for 1 min, and 90–5 % solvent B for 3 min.

The acquisition parameters on the Synapt G2 instrument were: source voltages—capillary (3.0 kV), sampling cone (40.0 V), and extraction cone (3.6 V); temperatures—source (100 °C) and desolvation (300 °C); MCP detector voltage (V).

MassLynx (Waters) was used for data processing with MaxEnt1 using the following settings: The resolution was set for 1 Da/channel for a uniform Gaussian model with minimum intensity ratios of 33% to left and right. The algorithm iterated until convergence.

### 4.6. Antibody Production

Vaccines were prepared essentially as described [56]. Briefly, vaccines were made by absorbing a carrier protein (ovalbumin) to an adjuvant, Al(OH)_3_, and pre-activating the carrier protein with IAANHOSu, which allows selective conjugation of peptides through -SH groups of the peptide Cys. The peptides used in the vaccines were three C-terminal peptides from mutated CRTfs (Table 4). Peptides used for immunisation were synthesized with an N-terminal Cys to assure conjugation to the carrier protein. The vaccines were controlled by amino acid analysis to ensure that the peptides were attached to the carrier. All vaccines had successfully attached peptides (data not shown).

Peptide antibodies were generated using traditional protocols [58,59]. Four mice were immunised per peptide. The mice were immunised every 14th day and bled 10 days after every immunisation. Four bleeds were taken from the immunised mice. The fourth bleed from each animal was analysed by ELISA. Following immunisation, spleen cells were harvested and fused with a standard hybridoma cell line (X63.Ag8.653) using conventional technology [58,59]. Final selection of reactive clones was conducted using biotinylated peptide 2 in a traditional ELISA.

### 4.7. Testing of Antibody Reactivity to Biotinylated Peptides by Enzyme-Linked Immunosorbent Assay

Microtiter plates were pre-coated with streptavidin (1 mg/mL) overnight at 4 °C in carbonate buffer, whereafter wells were rinsed with TTN (250 µL/well) and blocked with TTN for 1h at RT. Biotinylated peptides (1 μg/mL) were added to all wells and incubated for 1 h at RT. Between blocking and incubation with SSI-HYBs or goat anti-mouse IgG, wells were rinsed with TTN 3 × 1 min on a shaking table. Wells were incubated with SSI-HYBs (1:10 dilution or a 2-fold dilution series) for 1 h at RT and secondary antibody (goat anti-mouse IgG (1:2000)) for 1 h at RT as well. Bound antibodies were detected by adding AP substrate buffer along with *p*NPP (1 mg/mL, 100 µL/well) to all wells and reading at A_405-650_ on a microtiter plate reader after an appropriate colour development.

### 4.8. Testing of Antibody Reactivity to Resin-Bound Peptides by Modified Enzyme-Linked Immunosorbent Assay

Resin-bound peptides were added to a 96-well multiscreen filter plate (Millipore, Copenhagen, Denmark) and rinsed with TTN buffer. All incubations with antibodies diluted in TTN (1:10 for cell-culture supernatants, 1:1000 for purified antibodies) were carried out for 1 h at RT followed by three washes in TTN buffer. Resin beads were washed with TTN buffer using a multiscreen vacuum manifold. AP-conjugated goat anti-mouse IgG was used as secondary antibody. Bound antibodies were quantified using *p*NPP (1 mg/mL) diluted in AP substrate buffer. After a sufficient colour reaction, the buffer was transferred to a Maxisorp microtiter plate (Nunc, Roskilde, Denmark), and the absorbance was measured at 405 nm, with background subtraction at 650 nm, on a Thermo max microtiter plate reader (Molecular Devices, CA, USA).

### 4.9. Western Immunoblotting

Ten µg recombinant CRT/CRTfs L367/CRT K385 was diluted 1:2 in (non)-reducing sample buffer (SB) (+/−DTT) and incubated for 2 min at 95 °C. Twenty uL of each sample was loaded to wells of a 4–20% Tris glycine gel (Novex Life Technologies, CA, USA) and run for 2 h at 50 V and 250 mA and, later, 30 min at 100 V and 250 mA using Tris glycine SDS running buffer. Gels were stained overnight at 4 °C with Coomassie Brilliant Blue and washed with milli-Q water 5 times until bands were visualized.

For Western immunoblotting, the procedure was repeated as described above. After electrophoresis, gels were blotted onto a PVDF membrane using an iBlotter (Invitrogen Thermo Fisher, MA, USA). The membranes were blocked ON in TTN buffer and mounted in a blotting device. SSI-HYB 385-06 (diluted 1:1000) was added to each well and incubated for 1 h at RT. Washes were performed for 2 × 5 min. AP-conjugated goat anti-mouse IgG (1:1000) was incubated for 1 h on the membrane followed by washing 3 × 5 min. Finally, AP substrate (BCIP 0.5 mg/mL, NTB 0.3 mg/mL) was added and incubated until visible staining. The reaction was stopped by washing the membrane in milliQ water, followed by drying on filter paper. Finally, CRT FMC 75 mAb (diluted 1:10,000) was added to the membrane, and the bands were visualized as described above.

For Western blotting of MARIMO cells, the cells were cultured in RPMI medium together with 10% FBS and 1% PS media. MARIMO cells were either treated with methanol (96%) or BFA (1 µg/mL) for 7 h. Western blotting was conducted as already described, except that 20 µg denatured and reduced lysate was loaded per sample. Staining was conducted overnight at 4 °C using SSI-HYB 385-06 (1:2000) and CRT FMC 75 mAb (1:5000), whereafter bands were visualized using horseradish peroxidase-catalysed chemiluminescence (peroxidase AffiniPure goat anti-rabbit IgG or goat anti-mouse IgG) and quantified using ImageStudio (LI-COR Biosciences). The experiments were conducted in duplicates. 

### 4.10. Detection of Cells Positive for Frameshifted Calreticulin by Flow Cytometry

One mL full blood was used for each sample. All incubations were performed at RT, and all centrifugations were conducted at 500 g for 5 min. Erythrocytes were initially lysed by incubation with FACS Lysing solution for 10 min following centrifugation and decantation of supernatant. Next, the lymphocytes were permeabilized by incubation with permeabilizing solution (0.1% Tween 20, 0.5% paraformaldehyde in PBS) for 10 min and subsequent addition of washing solution (5% filtrated calf serum, 0.5% BSA, 0.07% sodium azide). After centrifugation, the supernatant was decanted, and washing solution was added, followed by another centrifugation and decantation of supernatant. Lymphocytes were subsequently stained with SSI-HYB 385-06 or mouse anti-human β-galactosidase IgG (dilution 1:1000), followed by a secondary staining with a FITC-labelled goat anti-mouse IgG antibody (dilution 1:1000). After staining for 1 h in the dark, leukocytes were washed twice in washing solution and then kept in fixation solution (1% paraformaldehyde in PBS) at 4 °C overnight. Flow cytometer analyses were performed by FACSCalibur flow cytometer and CELLQuest software (BD Biosciences, Franklin Lakes, NJ, USA). A total of 10,000 events were analysed.

## Figures and Tables

**Figure 1 ijms-23-06803-f001:**
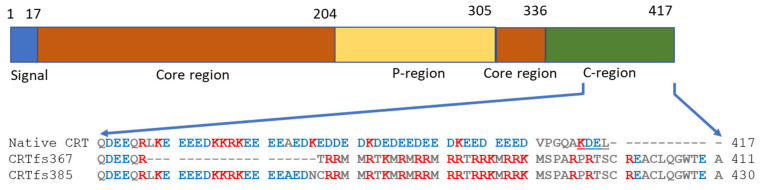
Schematic presentation of calreticulin (CRT) and location of C-terminal frameshift substitutions. The different regions are coloured according to Boelt et al. [13], and the C-terminal part from residue 361 of native CRT, CRTfs L367, and CRTfs K385 are highlighted (bottom). Acidic residues are coloured blue while basic residues are coloured red. The KDEL sequence of native CRT is underlined.

**Figure 2 ijms-23-06803-f002:**
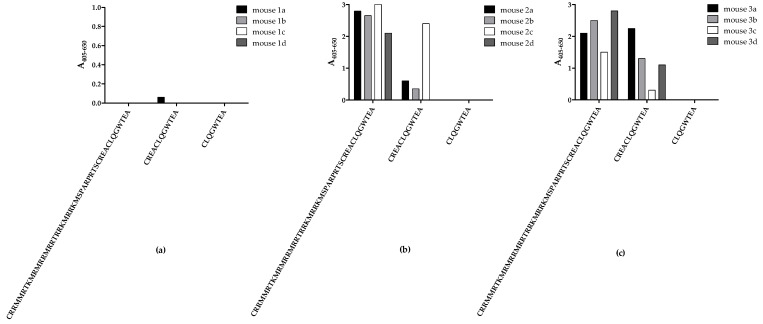
Reactivity of mouse bleeds to peptides used for immunisation tested in enzyme-linked immunosorbent assay. Fourth collection of mouse bleeds were tested for antibody reactivity to peptides 1–3: (**a**) Four mice (1a–d) were immunised with peptide 1 (CRRMMRTKMRMRRMRRTRRKMRRKMSPARPRTSCREACLQGWTEA), and the collected samples were tested for reactivity to peptides 1–3. (**b**) Four mice (2a–d) were immunised with peptide 2 (CREACLQGWTEA), and bleeds were tested for reactivity to peptides 1–3. (**c**) Four mice (3a–d) were immunised with peptide 3 (CLQGWTEA), and bleeds were tested for reactivity to peptides 1–3. Absorbances were corrected for background reactivity by subtracting reactivity from non-coated wells.

**Figure 3 ijms-23-06803-f003:**
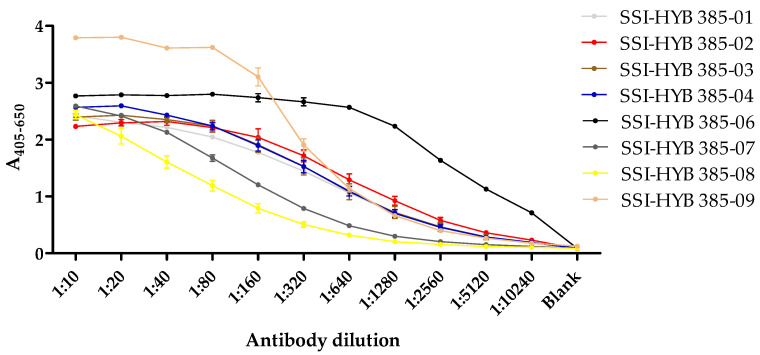
Titration of SSI-HYBs analysed by enzyme-linked immunosorbent assay. A two-fold dilution series ranging from 1:10–1:10240 was tested.

**Figure 4 ijms-23-06803-f004:**
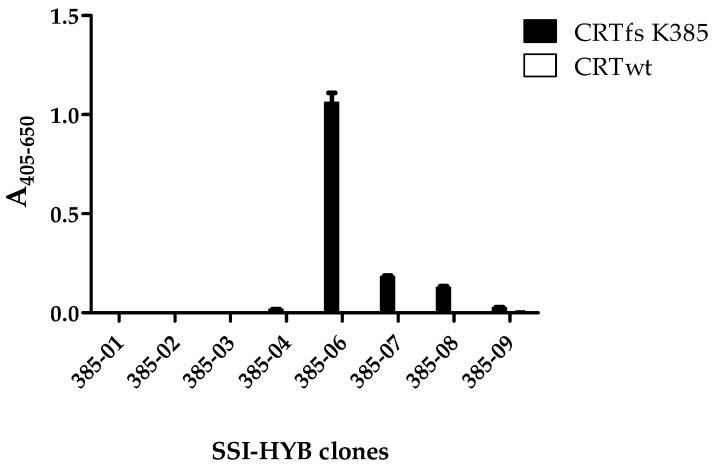
Reactivity of SSI-HYB clones to full-length frameshifted calreticulin (CRTfs) K385 and CRT wild-type (wt) analysed by enzyme-linked immunosorbent assay. Absorbances were corrected for background reactivity by subtracting reactivity from non-coated wells.

**Figure 5 ijms-23-06803-f005:**
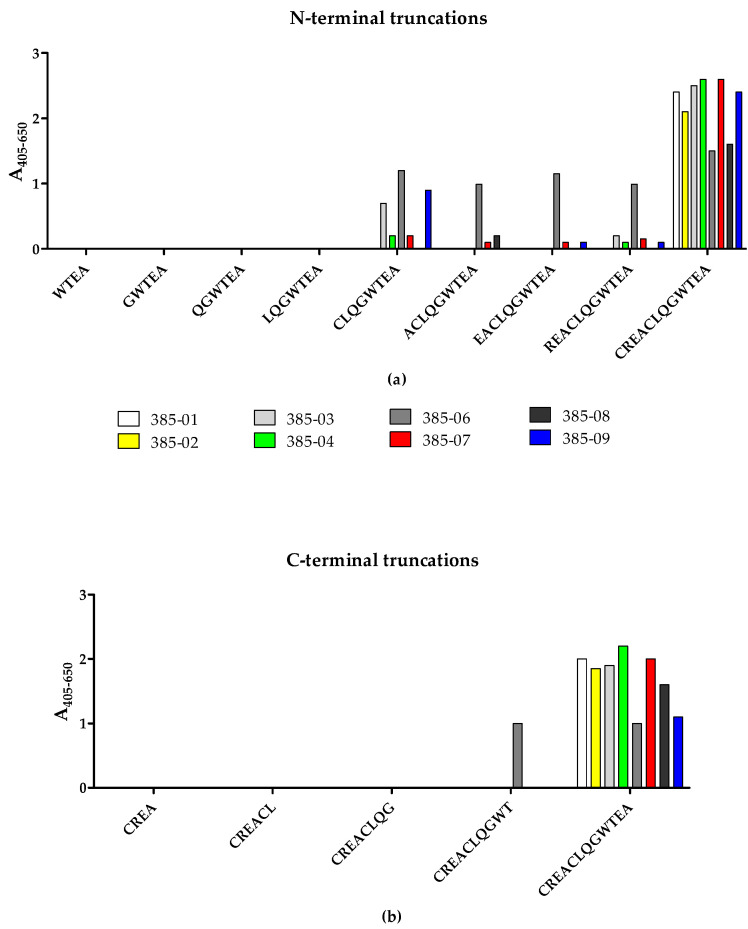
Reactivity of SSI-HYB clones to N- and C-terminally truncated frameshifted calreticulin peptides by modified enzyme-linked immunosorbent assay: (**a**) Reactivity of clones to N-terminally truncated resin-bound peptides. (**b**) Reactivity of clones to C-terminally truncated resin-bound peptides. Absorbances were corrected for background reactivity by subtracting reactivity from non-coated wells.

**Figure 6 ijms-23-06803-f006:**
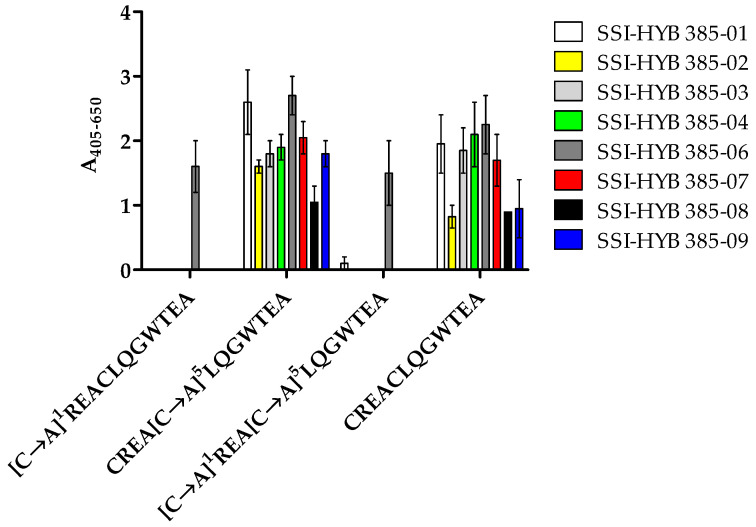
Reactivity of SSI-HYBs to Ala-substituted peptides analysed by modified enzyme-linked immunosorbent assay. The control peptide CREACLQGWTEA was used for generation of substituted peptides where Cys^1^ and Cys^5^ were substituted with Ala, and a single peptide where both Cys residues were substituted with Ala. Absorbances were corrected for background reactivity by subtracting reactivity from non-coated wells.

**Figure 7 ijms-23-06803-f007:**
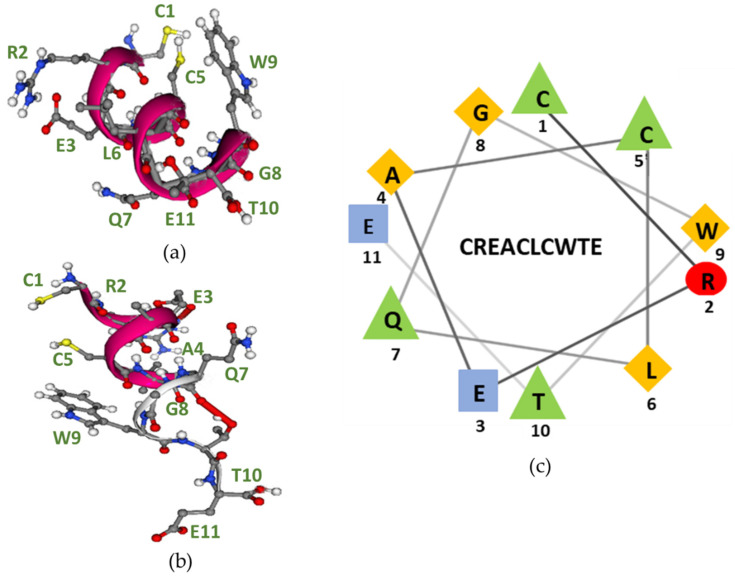
Structural epitope analysis. (**a**) The three-dimensional structure of the CREACLQGWTE sequence obtained from PEP FOLD SERVER. The peptide folds into an α-helix structure in which protruding amino acid residues are seen. On the right side of the structure, the side-chain residues of Trp^9^, Cys^1^, and Cys^5^ protrude. On the left side of the structure, the side-chain residues of Arg^2^, Glu^3^, Leu^6^, Gln^7^, and the terminal Thr^10^ protrude. (**b**) The three-dimensional structure of the CREACLQGWTE peptide. Yellow balls = sulphur atoms, red balls = oxygen atoms, grey balls = carbon atoms, white balls = hydrogen atoms, and blue balls = nitrogen atoms. (**c**) Helical wheel representation of the CREACLQGWTE peptide. The residues in the epitope are presented using one-letter codes. The image was generated using the NetWheels application (http://lbqp.unb.br/NetWheels, accessed on 14 May 2022). The following colours represent amino acid functions: red, polar/basic; blue, polar/acid; green, polar/uncharged; yellow, nonpolar.

**Figure 8 ijms-23-06803-f008:**
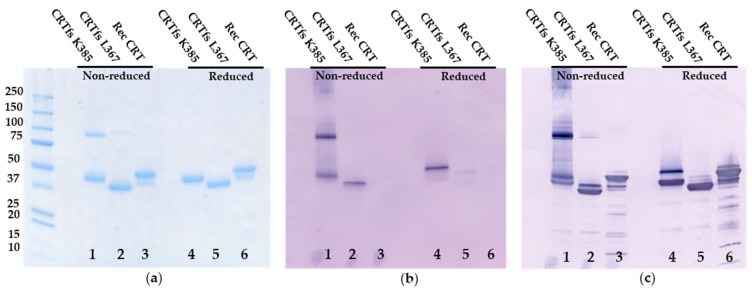
Immunostaining of SSI-HYB 385-06 reactivity to frameshifted calreticulin (CRTfs) and wild-type calreticulin (CRTwt). (**a**) Coomassie Brilliant Blue staining of recombinant CRT proteins. (**b**) Western blotting of SSI-HYB 385-06 reactivity to CRTfs L367 and K385 and CRTwt. (**c**) Western blotting of membrane from figure (**b**) developed with a commercial monoclonal antibody (CRT FMC 75 mAb), recognizing amino acids 34–41 (TSRWIESK) in the N-terminal of CRT. Lane 1: non-reduced CRTfs K385, Lane 2: non-reduced CRTfs L367, Lane 3: non-reduced recombinant CRT, Lane 4: reduced CRTfs K385, Lane 5: reduced CRTfs L367, Lane 6: reduced recombinant CRT.

**Figure 9 ijms-23-06803-f009:**
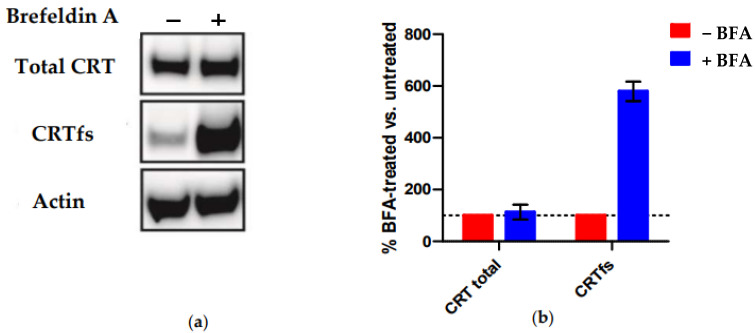
Reactivity of SSI-HYB 385-06 to frameshifted calreticulin (CRTfs) in MARIMO cells analysed by Western blotting. (**a**) MARIMO wells were treated with methanol (−brefeldin A (BFA)) or with BFA, whereafter cells were lysed and examined by Western blotting under reduced conditions. The monoclonal antibody FMC-75 to calreticulin (CRT) was used to detect the total intracellular CRT level, as this antibody interacts with an epitope in the N-terminal, thus detecting wildtype and mutated CRT [37]. CRTfs was stained using SSI-HYB 385-06. Actin was used as loading control (**b**). Quantification of CRT and CRTfs levels based on Western blot in (**a**).

**Figure 10 ijms-23-06803-f010:**
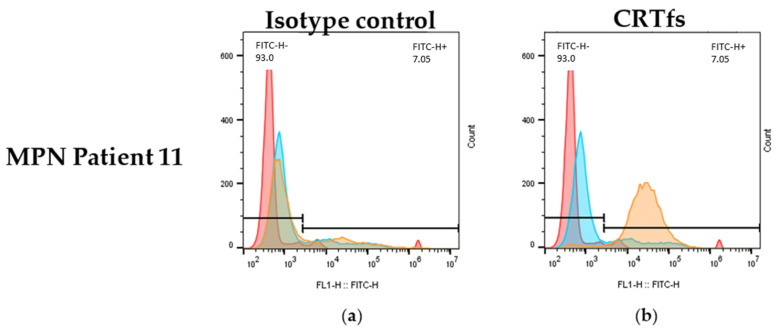
Intracellular leukocyte staining of frameshifted calreticulin (CRTfs) by flow cytometry. (**a**) Negative isotype control staining using a mouse monoclonal antibody of irrelevant specificity (β-galactosidase). (**b**) Staining using SSI-HYB 385-06 recognizing CRTfs. Colours represent: red, unstained cells; blue, non-specific FITC staining; orange, specific CRT staining.

**Table 1 ijms-23-06803-t001:** Antibody clones selected for final characterization.

Name	Clone Number	Subclass	Epitope
SSI-HYB 385-01	SSI-5F7	IgG1/kappa	CREACLQGWTE
SSI-HYB 385-02	SSI-9B2	IgG1/kappa	CREACLQGWTE
SSI-HYB 385-03	SSI-9B8	IgG1/kappa	CREACLQGWTE
SSI-HYB 385-04	SSI-11E5	IgG1/kappa	CREACLQGWTE
SSI-HYB 385-06	SSI-4F10	IgG1/kappa	CLQGWT
SSI-HYB 385-07	SSI-8F10	IgG1/kappa	CREACLQGWTE
SSI-HYB 385-08	SSI-11G2	IgG1/kappa	CREACLQGWTE
SSI-HYB 385-09	SSI-1F12	IgG1/kappa	CREACLQGWTE

**Table 2 ijms-23-06803-t002:** SSI-HYB reactivity to Ala-substituted peptides. Amino acids in brackets represent the amino acid substituted by Ala. N represent the number of substitutions that were not tolerated in relation to obtaining antibody binding. The peptide CREAALQGWTEA was used for generation of substituted peptides. ++ represents reactivity similar to the control, + represents medium reactivity, − represents no reactivity.

	[R]^2^	[E]^3^	[L]^6^	[Q]^7^	[G]^8^	[W]^9^	[T]^10^	[E]^11^	N
SSI-HYB 385-01	+	+	+	++	++	−	−	−	3
SSI-HYB 385-02	+	−	−	+	−	−	−	++	5
SSI-HYB 385-03	++	++	−	+	−	−	−	++	4
SSI-HYB 385-04	++	++	−	+	−	−	−	++	4
SSI-HYB 385-06	++	++	++	++	++	−	++	++	1
SSI-HYB 385-07	++	++	−	++	++	−	++	−	3
SSI-HYB 385-08	+	++	−	++	++	−	++	−	3
SSI-HYB 385-09	++	++	−	++	++	−	++	−	3

**Table 3 ijms-23-06803-t003:** SSI-HYB reactivity to functionality-substituted peptides. The amino acids in brackets represent the substitution introduced. The peptide CREAALQGWTEA was used for generation of substituted peptides. ++ represents reactivity similar to the control, + represents medium reactivity, − represents no reactivity.

	[C→M]^1^	[A→G]^4^	[L→I]^6^	[W→F]^9^	[T→S]^10^	[E→D]^11^	[A→G]^12^
SSI-HYB 385-01	−	++	++	++	+	−	++
SSI-HYB 385-02	−	++	++	−	++	++	++
SSI-HYB 385-03	−	++	++	++	++	++	++
SSI-HYB 385-04	−	++	++	−	++	++	++
SSI-HYB 385-06	++	++	++	++	++	++	++
SSI-HYB 385-07	−	+	+	++	++	−	++
SSI-HYB 385-08	−	+	++	++	++	−	++
SSI-HYB 385-09	−	++	++	++	++	−	++

**Table 4 ijms-23-06803-t004:** Synthetic peptides used for immunisation. Peptides were synthesized with an N-terminal Cys to assure conjugation to the carrier.

Peptide	Sequences
1	(C)RRMMRTKMRMRRMRRTRRKMRRKMSPARPRTSCREACLQGWTEA
2	(C)REACLQGWTEA
3	(C)LQGWTEA

**Table 5 ijms-23-06803-t005:** Synthetic peptides used for characterization of peptide antibodies. Amino acids underlined represent substituted amino acids.

N-Terminally Truncated Peptides	C-Terminally Truncated Peptides	Ala-Substituted Peptides	Functionality-Substituted Peptides
CREACLQGWTEA	CREACLQGWTEA	AREACLQGWTEA	MREACLQGWTEA
REACLQGWTEA	CREACLQGWT	CAEACLQGWTEA	CREGCLQGWTEA
EACLQGWTEA	CREACLQG	CRAACLQGWTEA	CREACIQGWTEA
ACLQGWTEA	CREACL	CREAALQGWTEA	CREACLQGFTEA
CLQGWTEA	CREA	CREACAQGWTEA	CREACLQGWSEA
LQGWTEA		CREACLAGWTEA	CREACLQGWTDA
QGWTEA		CREACLQGATEA	CREACLQGWTEG
GWTEA		CREACLQGWAEA	
WTEA		CREACLQGWTAA	

## Data Availability

Not applicable.

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
