# Peer review of "Production and Characterization of Peptide Antibodies to the C-Terminal of Frameshifted Calreticulin Associated with Myeloproliferative Diseases"

_ijms, 2022, doi:10.3390/ijms23126803_

Round 1
Reviewer 1 Report
This manuscript by Mughal et al describes the development and characterization of novel antibodies raised against the mutant forms of Calreticulin (CRT) found in myeloproliferative neoplasms (MPNs). Mutant CRT in MPNs function to activate signaling by the thrombopoietin receptor (encoded by MPL) thereby inducing a disease-driving signal that is based on activation of the JAK2 tyrosine kinase. Mutant CRT protein are produced following insertions/deletions at sequences encoding the c-terminus of CRT, resulting in frameshifts that lead to a unique c-terminal sequence. Thus, an antibody that specifically recognizes mutant CRT has potential value as research tools as well as potential therapeutics for MPN. The manuscript is well written and for the most part quite clear. The main concern is the applicability of the antibodies produced, and as such, the significance of the work to the MPN field is unclear. Addressing the following concerns will improve the manuscript.
Major
- The main concern is the lack of data showing antibody reactivity with CRTfs expressed in cells, specifically cells from CRT mutant MPN patients. Such data is mentioned in the results, but these data are not shown. In fact, these data are referenced in the abstract, which is misleading, considering the data are not shown in the manuscript and therefore aren’t peer-reviewed.
- The results would be more significant/impressive if the antibodies can detect mutant CRT expressed in mammalian cells and not simply purified protein expressed in bacteria (was the protein source indicated?).
- On line 75 the authors state “Despite the knowledge of the genetic basics of these diseases, MPN therapies have not yet exploited the knowledge of mutations in treatment of MPNs, which could selectively target the malignant clones [14].” The JAK2 tyrosine kinase is mutationally activated in MPN and JAK2 inhibitors are approved and lead to improvements in symptoms and quality of life, can improve survival, and can improve mutant allele burden (suggesting potential targeting of malignant clone). While these drugs are not specific to mutation induced signaling, and may not readily select malignant clones, they have been developed based on the mutations in MPNs and many patients are benefitting from them. This section should be edited to acknowledge MPN patients have benefited from JAK2 inhibitors.
- The authors should provide a description as to why antibodies for the use of IHC in MPN would be beneficial compared to genomic identification of the CRT mutations.
- On line 160 the authors state the minimum functional epitope of 385-06 is CLQGWT but it is unclear why this is concluded. It is understood that CLQGWTEA binds to the antibody, and CREACLQGWT binds to the antibody, but how is it concluded that CLQGWT is the minimum functional epitope? With that said, the authors state on Line 249 that CLQGW is minimal for 385-06. The authors should clarify the consensus and the reasoning for their defining it as the consensus.
- It is unclear why CREAALQGWTEA was used as the basis for mutagenesis and not the actual peptide, CREACLQGWTEA, present in the antigen.
- Given the need for certain backbone functional, but not specific, residues in the consensus binding sites of the antibodies, can the authors comment on the presence of the consensus sequence or similar sequences in any other proteins encoded by the human genome? The antibodies need to be fairly specific to have value in an IHC assay or a therapeutic.
- In line 340 the authors indicate the data suggest a cleaved c-terminus and its binding back to the rest of the protein. Can the authors explain the evidence for this better for the reader?
- While the authors didn’t show detection of CRTfs expressed in cells by their novel antibodies, can the authors comment on the extent to which their data may predict the usefulness of their antibodies in IHC or flow cytometry, where the native CRTfs proteins would be complexed with the MPL protein? Would the consenus binding sequence be accessible to an antibody while CRTfs protein is complexed with MPL protein?
Minor
- In Figure 2 it would be helpful to indicate that 1a, 1b, 1c, and 1d, etc. are referring to different mice.
- It would be helpful to include the peptides (aligned) in Figure 2, and not just in the figure legend.
- In Figure 5 it would be helpful if colors were used instead of shades of grey.
- Do the authors mean [C-A]1 and not [C-A]5 in line 173?
- In Figure 6 it would be helpful to underline the mutations in the peptides on the x-axis, or make the mutated residues a different color to make the changes more readily apparent to the reader.
- In Figure 6a the amino acids and their numbers are hard to read and need to be shifted and perhaps colored in a color absent in the structure.
- In Lines 247 and 248 it isn’t clear why G-8 is not included in the consensus peptides. Is this a typo?
- In Figure 7 the gel lanes should be labeled at the top. It is difficult for the reader to refer to the legend to understand which sample is in each lane.
- In line 423 the authors indicate the Trp-Thr motif may work cooperatively in the epitope, because that has been shown before for the Thr-Trp motif. This seems rather speculative in nature.
Author Response
Dear reviewer
Thank you for the constructive comments to the CRT Ab manuscript raised by you. We have addressed all the comments and your points have improved the manuscript. Each comment is addressed below.
Best regards
Gunnar Houen and Nicole Trier
Comments and Suggestions for Authors
This manuscript by Mughal et al describes the development and characterization of novel antibodies raised against the mutant forms of Calreticulin (CRT) found in myeloproliferative neoplasms (MPNs). Mutant CRT in MPNs function to activate signaling by the thrombopoietin receptor (encoded by MPL) thereby inducing a disease-driving signal that is based on activation of the JAK2 tyrosine kinase. Mutant CRT protein are produced following insertions/deletions at sequences encoding the c-terminus of CRT, resulting in frameshifts that lead to a unique c-terminal sequence. Thus, an antibody that specifically recognizes mutant CRT has potential value as research tools as well as potential therapeutics for MPN. The manuscript is well written and for the most part quite clear. The main concern is the applicability of the antibodies produced, and as such, the significance of the work to the MPN field is unclear. Addressing the following concerns will improve the manuscript.
Major
- The main concern is the lack of data showing antibody reactivity with CRTfs expressed in cells, specifically cells from CRT mutant MPN patients. Such data is mentioned in the results, but these data are not shown. In fact, these data are referenced in the abstract, which is misleading, considering the data are not shown in the manuscript and therefore aren’t peer-reviewed.
Response: We agree that the abstract is misleading as mentioned results are not illustrated in the text, as a consequence the abstract was rephrased: Moreover, a new figure was added, to illustrate the reactivity of SSI-HYB 385-06 to frameshifted CRT in MPN patients by flow cytometry. Moreover, a new experiment was introduced, illustrating the reactivity of SSI-HYB 385-06 to MARIMO cells, expressing frameshifted CRT L367.
- The results would be more significant/impressive if the antibodies can detect mutant CRT expressed in mammalian cells and not simply purified protein expressed in bacteria (was the protein source indicated?).
Response: CRTfs was originally expressed in yeast, which has been added where relevant. Moreover, an experiment testing the reactivity of SSI-HYB 385-06 to MARIMO cells expressing CFRfs L367 was included and an additional figure (Figure 8), showing the reactivity of SSI-HYB 385-06 to white blood cells of a MPN patient was included as well (Figure 9).
- On line 75 the authors state “Despite the knowledge of the genetic basics of these diseases, MPN therapies have not yet exploited the knowledge of mutations in treatment of MPNs, which could selectively target the malignant clones [14].” The JAK2 tyrosine kinase is mutationally activated in MPN and JAK2 inhibitors are approved and lead to improvements in symptoms and quality of life, can improve survival, and can improve mutant allele burden (suggesting potential targeting of malignant clone). While these drugs are not specific to mutation induced signaling, and may not readily select malignant clones, they have been developed based on the mutations in MPNs and many patients are benefitting from them. This section should be edited to acknowledge MPN patients have benefited from JAK2 inhibitors.
Response: JAK2 inhibitors and other potential disease-modifying drugs have been mentioned in the introduction.
- The authors should provide a description as to why antibodies for the use of IHC in MPN would be beneficial compared to genomic identification of the CRT mutations.
Response: This has been elaborated in the introduction. In our belief the 2 methods supplement each other.
- On line 160 the authors state the minimum functional epitope of 385-06 is CLQGWT but it is unclear why this is concluded. It is understood that CLQGWTEA binds to the antibody, and CREACLQGWT binds to the antibody, but how is it concluded that CLQGWT is the minimum functional epitope? With that said, the authors state on Line 249 that CLQGW is minimal for 385-06. The authors should clarify the consensus and the reasoning for their defining it as the consensus.
Response: The minimum functional epitope is defined as the shortest possible peptide necessary for antibody reactivity. Based on screenings of N-terminal truncated peptides, reactivity was found to the CLQGWTEA peptide. Based on screening of C-terminal truncated peptides, the amino acids -EA were not observed to be essential for reactivity. Based on these results, it was assumed that the shortest sequence for antibody binding is C5-T10, although this remains to be finally verified. We acknowledge that the formulation can be misleading, and the paragraphs has been rephrased as the epitope of 385-06 was not finally confirmed.
- It is unclear why CREAALQGWTEA was used as the basis for mutagenesis and not the actual peptide, CREACLQGWTEA, present in the antigen.
Response: The peptide CREAALQGWTEA was used for testing of substitutions to avoid obstacles with the possible formation of a disulfide bond between the two Cys residues. This has been elaborated in the text.
- Given the need for certain backbone functional, but not specific, residues in the consensus binding sites of the antibodies, can the authors comment on the presence of the consensus sequence or similar sequences in any other proteins encoded by the human genome? The antibodies need to be fairly specific to have value in an IHC assay or a therapeutic.
Response: A protein blast search was conducted, which found a sequence match for the 6 amino acids CLQGWT in NOTCH4. However, since NOTCH4 has a molecular weight of approximately 200000 Da, is a transmembrane protein with a tissue and cellular distribution, we do not believe that this will interfere with the use of this antibody but it has to be kept in mind when using this antibody. Moreover, the identified motif 870CLQGWT870 is proposed to be located in a flexible turn structure in NOTCH4, when conducting structural analysis using Alphafold, which is different from the proposed α-helix structure identified in CRTfs, supporting that the SSI-HYB 385-06 does not interact with NOTCH4, due to structural differences. This has been elaborated in the discussion section.
- In line 340 the authors indicate the data suggest a cleaved c-terminus and its binding back to the rest of the protein. Can the authors explain the evidence for this better for the reader?
Response: The paragraph has been rephrased.
- While the authors didn’t show detection of CRTfs expressed in cells by their novel antibodies, can the authors comment on the extent to which their data may predict the usefulness of their antibodies in IHC or flow cytometry, where the native CRTfs proteins would be complexed with the MPL protein? Would the consenus binding sequence be accessible to an antibody while CRTfs protein is complexed with MPL protein?
Response: Studies of truncated mutants have revealed that residues 376-383 from the C-terminus of CRTDel52 are required to activate MPL-mediated signaling. Recent studies indicate that the C-terminal Cysteines are essential for reactivity as well, as mutations C400A/C404A in CRTDel52 reduced its ability to bind MPL, however reactivity was still obtained. Based on these findings our current belief is that the SSI-HYB 385-06, which does not have a dependency of any of the C-terminal Cys residues would be able to interact with CRTfs in complex with MPL. This has been elaborated in the text.
Minor
- In Figure 2 it would be helpful to indicate that 1a, 1b, 1c, and 1d, etc. are referring to different mice.
Response: the explanation 1-d etc were originally mentioned in the figure legend, but has been added to the figure as well.
- It would be helpful to include the peptides (aligned) in Figure 2, and not just in the figure legend
Response: the peptide sequences have been written in the figure instead of the numbers
- In Figure 5 it would be helpful if colors were used instead of shades of grey.
Response: a new Figure 5 using colors has been added.
- Do the authors mean [C-A]1 and not [C-A]5 in line 173?
Response: Yes, thank you for noticing. Corrected.
- In Figure 6 it would be helpful to underline the mutations in the peptides on the x-axis, or make the mutated residues a different color to make the changes more readily apparent to the reader.
Response: Figure 6 has been modified to clarify the substitutional changes and was color-matched according to Figure 5.
- In Figure 6a the amino acids and their numbers are hard to read and need to be shifted and perhaps colored in a color absent in the structure.
Response: The figure has been modified according to the suggestions.
- In Lines 247 and 248 it isn’t clear why G-8 is not included in the consensus peptides. Is this a typo?
Response: It is correct that only the [G→A]8-substituted peptides were tested for reactivity. As Gly is a small amino acid, which only contains a hydrogen bond in the side chain it is challenging to substitute with a relevant amino acid, thus substitution of Gly with Ala functions as a functionality substitution. This was elaborated in the text.
- In Figure 7 the gel lanes should be labeled at the top. It is difficult for the reader to refer to the legend to understand which sample is in each lane.
Response: Figure 7 has been modified to make it more readable.
- In line 423 the authors indicate the Trp-Thr motif may work cooperatively in the epitope, because that has been shown before for the Thr-Trp motif. This seems rather speculative in nature.
Response: We acknowledge that the HYB that appeared to have a dependency for the amino acids upon Ala scanning, did not depend on the complete motif when screening functionality substituted peptides. As a consequence, the paragraph was removed.
Reviewer 2 Report
I think it is important to give space to this manuscript, because the identification of antibodies capable of targeting CRTfs can have numerous diagnostic and therapeutic implications in MPNs.
The article is very technical and on the basis of my expertise it presents many interesting points and no methodological problems.
Some suggestions below:
Line 72. Therapies are only partially listed. Target drugs such as JAK inhibitors and potential "disease modifiers" currently available in trials (imetelstat, navtemadlin, pelabresib etc) are omitted.
The article also presents two problems of structure:
- it should not close with the description of materials and methods, which should anticipate the results
- it presents a scarce discussion about the potential applications of the findings in clinical hematology, and it would be interesting to obtain a speculation from the authors on this subject
Author Response
Dear reviewer
Thank you for the constructive comments to the CRT Ab manuscript raised by you. We have addressed all the comments and your points have improved the manuscript. Each comment is addressed below.
Best regards
Gunnar Houen and Nicole Trier
Comments and Suggestions for Authors
I think it is important to give space to this manuscript, because the identification of antibodies capable of targeting CRTfs can have numerous diagnostic and therapeutic implications in MPNs.
The article is very technical and on the basis of my expertise it presents many interesting points and no methodological problems.
Some suggestions below:
Line 72. Therapies are only partially listed. Target drugs such as JAK inhibitors and potential "disease modifiers" currently available in trials (imetelstat, navtemadlin, pelabresib etc) are omitted.
Response: JAK inhibitors and disease modifiers have been added to the introduction.
The article also presents two problems of structure:
- it should not close with the description of materials and methods, which should anticipate the results
- it presents a scarce discussion about the potential applications of the findings in clinical hematology, and it would be interesting to obtain a speculation from the authors on this subject
Response: We agree that the materials and methods section may be misplaced. However, the manuscript follows the format guidelines presented by the journal, which states that the material and methods section is presented immediately after the results and discussion section.
The presented results have been discussed in relation to the potential applications of the findings in clinical hematology.